# Attention-Based Reward Shaping for Sparse and Delayed Rewards

## Abstract

Sparse and delayed reward functions pose a significant obstacle for real-world Reinforcement Learning (RL) applications. In this work, we propose *Attention-based REward Shaping (ARES)*, a general and robust algorithm which uses a transformer's attention mechanism to generate shaped rewards and create a dense reward function for any environment. ARES requires a set of episodes and their final returns as input. It can be trained entirely offline and is able to generate meaningful shaped rewards even when using small datasets or episodes produced by agents taking random actions. ARES is compatible with any RL algorithm and can handle any level of reward sparsity. In our experiments, we focus on the most challenging case where rewards are fully delayed until the end of each episode. We evaluate ARES across a diverse range of environments, widely used RL algorithms, and baseline methods to assess the effectiveness of the shaped rewards it produces. Our results show that ARES can indeed improve learning in delayed reward settings, enabling RL agents to train in scenarios that would otherwise require impractical amounts of data or even be unlearnable, though there remain some cases where ARES is not successful in doing so. To our knowledge, ARES is the first approach that works fully offline, remains robust to extreme reward delays and low-quality data, and is not limited to goal-based tasks.

**Link to anonymized GitHub repository**

## 1 Introduction

Reinforcement Learning (RL) has been successfully applied to a wide range of problems, such as robotics (M. Andrychowicz & Zaremba., 2019), strategy games (Berner et al., 2019), and more recently training large language models to align with human preferences (RLHF) (L. Ouyang & Lowe., 2022). Yet many real-world RL applications are fundamentally limited by the challenge of dealing with *sparse* and *delayed* reward settings, where feedback signals are rare (sparse) or only provided at the end of an episode (delayed). Such sparse or delayed rewards dramatically increase the difficulty of temporal credit assignment, slowing learning and often preventing agents from discovering successful behaviors (Sutton & Barto., 2018). Addressing this challenge typically requires either manual designing of a dense function by a domain expert or large-scale exploration, both of which can be tedious, unreliable (J. Skalse & Krueger., 2022), or simply infeasible with complex environments (Sutton & Barto., 2018).

The most common algorithmic approach is to generate a set of rewards denser than those provided by the original environment, in a process called **reward shaping**. This is the approach taken by many recent algorithms such as Iterative Relative Credit Refinement (T. Gangwani & Peng., 2020), Randomized Return Decomposition (Z. Ren & Peng., 2022), Learning Online with Guidance Offline, (D. Rengarajan & Shakkottai., 2022), Dense reward learning from Stages (T. Mu & Su., 2024), Reinforcement Learning Optimising Shaping Algorithm (D. Mguni & Yang., 2023), Align-RUDDER (V. Patil & Hochreiter., 2022), and Attention-Based Credit (A. J. Chan & van der Schaar., 2024). In general, these methods take as input episodes from the environment of interest, and then output a set of rewards which is denser than that provided by the original environment. In the sparse or delayed reward case, the environment's returned rewards are mostly zero, and we want to replace these rewards with some more informative value.

We categorize these previous approaches under four categories: (1) whether the method is designed to address the delayed reward case (as opposed to only addressing the sparse reward case), (2) whether the method is designed to work offline (as opposed to only working online), (3) whether the method is designed to work with non-expert data (as opposed to only working with expert data), and (4) whether the method is designed for the general RL case (as opposed to only being usable in a particular RL setting, like goal-based RL). We find that all previously proposed algorithms, to the best of our knowledge, fail to meet at least one of the criteria outlined in Table 1, and thus, without substantial modification, are limited in their applicability to the broader set of RL use cases. A detailed summary of these limitations by algorithm is provided in the Related Work section.

In this work, we propose Attention-based REward Shaping (ARES), a method designed to be broadly applicable across RL settings. Given an offline dataset consisting of episodes from the target environment, each labeled with a final return, we first train a transformer model (A. Vaswani & Polosukhin., 2017) to predict the return associated with each episode. We then leverage the transformer's attention matrix to derive a reward signal for every timestep, associated with the corresponding state-action pair. These shaped rewards are subsequently used to train a RL agent on the original environment. The generality and effectiveness of ARES is evaluated on a diverse suite of environments and across different RL algorithms. All experiments are conducted under the most general and challenging conditions: delayed rewards, offline training, non-expert data, and standard RL settings. Our results show mixed success: we demonstrate that ARES can improve training performance and, in some cases, achieves performance comparable to the ideal scenario of immediate reward availability, but there are some settings where ARES fails, such as the HalfCheetah environment. Note that by "offline training" we mean training from a fixed dataset of collected experiences, i.e. a buffer of episodes collected from the environment of interest. ARES displays moderate invariance to the particular RL algorithm used: while absolute performance varies across algorithms (as it does with true immediate rewards), the relative improvement provided by ARES remains generally consistent.

To summarize, our contributions are as follows:

1. ARES combines the ability to use suboptimal episodes with the ability to work offline and with fully-delayed environments. In addition, it shows compatibility with a wide array of diverse RL algorithms, and it is not restricted to a goal-based RL setting. As a result, it is the most general algorithm to date for solving the sparse and delayed reward problems using shaping.

2. We evaluate and confirm this generality by demonstrating that ARES boosts training performance on a wide range of environments and RL algorithms, showing in addition to the above point that ARES is moderately invariant to the RL algorithm used. (Contribution 2 is justified purely by empirical results.)

## 2 Related Work

Table 1: Comparison of selected algorithms.

| Paper Title | Delayed? | Offline? | Non-expert data? | General? |
|---|---|---|---|---|
| Keeping Your Distance (Sibling Rivalry) (A. Trott & Socher., 2019) | Unclear | No | Yes | Goal-based |
| Learning Guidance Rewards with Trajectory-Space Smoothing (T. Gangwani & Peng., 2020) | No | No | Yes | Yes |
| DrS: Reusable Dense Rewards for Multi-Stage Tasks (T. Mu & Su., 2024) | No | Yes | Yes | Goal-based |

| | | | | |
|---|---|---|---|---|
| Attention Based Credit (ABC) (A. J. Chan & van der Schaar., 2024) | Yes | Yes | No | RLHF |
| Predictive Coding for Boosting Deep RL with Sparse Rewards (Preprint) (X. Lu & Abbeel., 2020) | Unclear | Yes | Yes | Goal-based |
| Randomized Return Decomposition (RRD) (Z. Ren & Peng., 2022) | No | No | Yes | Yes |
| Sequence Modeling of Temporal Credit Assignment for Episodic RL (Preprint) (Y. Liu & Peng., 2019) | No | No | Yes | Yes |
| Reinforcement Learning with Sparse Rewards Using Guidance from Offline Demonstration (LOGO) (D. Rengarajan & Shakkottai., 2022) | No | No | Yes | Yes |
| Self-Attentional Credit Assignment for Transfer in RL (SECRET) (J. Ferret & Pietquin., 2020) | Both | Yes | No | Yes |
| Synthetic Returns for Long-Term Credit Assignment (Preprint) (D. Raposo & Song., 2021) | Both | No | Yes | Yes |
| Self-Imitation Learning for Sparse and Delayed Rewards (Chen & Lin., 2021) | Both | No | Yes | Yes |
| Reward Design via Online Gradient Ascent (J. Sorg & Lewis., 2010) | Yes | No | Yes | Yes |
| T-REX (D. S. Brown & Niekum., 2019) | Yes | No | Yes | Yes |
| Self-Supervised Online Reward Shaping (SORS) (F. Memarian & Topcu., 2021) | Yes | No | Yes | Yes |
| Learning to Shape Rewards Using a Game of Two Partners (ROSA) (D. Mguni & Yang., 2023) | Yes | No | Yes | Yes |
| DARA: Dynamics-Aware Reward Augmentation in Offline RL (J. Liu & Wang., 2022) | Yes | Yes | No | Yes |
| ALIGN-RUDDER (V. Patil & Hochreiter., 2022) | Yes | Yes | No | Yes |

| Hindsight Experience Replay (HER) (M. Andrychowicz & Zaremba., 2017) | Yes | Yes | Yes | Goal-based |
|---|---|---|---|---|
| Attention Based REward Shaping (ARES) **(Proposed)** | Yes | Yes | Yes | Yes |

Table 1 shows a breakdown of each algorithm under the categories we described in the previous section: whether the algorithm addresses the case of fully delayed rewards, whether it works offline, whether it can use non-expert data, and whether it is applicable to the general RL setting. In Table 1, if "Delayed?" is "No," then the algorithm is only evaluated on the sparse case.

The algorithm **Attention-Based Credit** (A. J. Chan & van der Schaar., 2024) merits some discussion. It works in the context of large language models (LLMs) and RLHF, where the input is text made up of tokens. It uses the attention weights of an RLHF model, which is an attention-based delayed reward model that has been manually trained by humans, to redistribute those delayed rewards to the individual tokens that make up the input text. This produces a shaped reward signal which is derived from the attention matrix (though in a very different way from our approach) and does not require online interaction. Other than the name, ABC and ARES have a number of major differences. The two most major ones: ABC requires an "expert" reward model that has been trained by humans, while ARES works with trajectories of any quality and creates a new reward function. Also, the ARES algorithm is more complicated and relies on much more than the attention scores alone; the simplicity of the ABC algorithm means that it cannot, for example, handle an environment where there are both positive and negative rewards, because it uses the attention scores as a direct measure of credit.

Here we will briefly introduce *goal-based RL* and its limitations. In the goal-based or goal-conditioned RL setting, the reward function takes as a parameter (possibly the only parameter) whether or not some "goal" was achieved. For example, the task could be for a robot to push a box, and the goal is whether or not the box has been moved. This typically leads to a sparse or delayed reward function which is binary in nature: for example, at every returning 0 if the goal is not currently met and +1 only if it is. While this simplicity can be appropriate for some settings, there are many environments where there are not easily-defined goals that can be used to delineate success, and in general the binary reward function is by nature less rich than an ordinary RL reward function, which can output many values and provide a greater range of evaluation. Note that in some formulations, there can be subgoals which can be combined for a non-binary reward function; however, the general problems still remain, because it can be difficult to define subgoals, and each subgoal is itself still binary.

## 3 Problem Setting

Let $\mathcal{M} = (\mathcal{S}, \mathcal{A}, P, R, \mu)$ denote a Markov Decision Process (MDP), where $\mathcal{S}$ is the state space, $\mathcal{A}$ the action space, $P$ the transition distribution, $R$ the reward function, and $\mu$ the initial state distribution. We define the environment's original reward function as:

$$R_{\text{original}}(s \sim \mathcal{S}, \ a \sim \mathcal{A}) \tag{1}$$

An agent interacts with an environment by generating trajectories $\tau = (s_0, a_0, s_1, \ldots, s_T)$ over episodes of finite horizon $T$. The agent's behavior is governed by a policy $\pi(a \mid s)$, and the objective is to maximize the expected return by finding the optimal policy.

$$\pi^* = \arg\max_{\pi} \mathbb{E}_{\pi} \left[ \sum_{t=0}^{T} R_{\text{original}}(s_t, a_t) \right] \tag{2}$$

We now formally define the key types of reward functions:

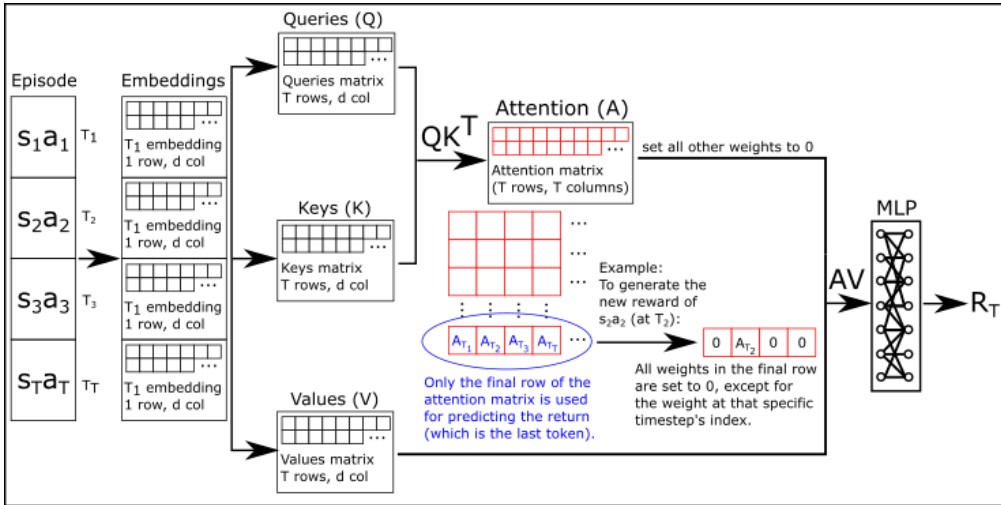

Figure 1: ARES Architecture.

**Delayed Reward Function:** A reward function is *delayed* if rewards are received only at the end of an episode. Formally, for an episode of length $T$:

$$R_{\text{delayed}}(s_t, a_t) = \begin{cases} 0 & \text{if } t < T \\ \sum_{i=0}^{T} R_{\text{original}}(s_i, a_i) & \text{if } t = T \end{cases} \tag{3}$$

**Sparse Reward Function:** A reward function is *sparse* if feedback signals are rare but not necessarily delayed to the end of the episode. That is, $R(s_t, a_t) = 0$ for most timesteps $t$, but may be nonzero at intermediate steps.

**Dense Reward Function:** A reward function is *dense* if rewards are immediate and provided at every timestep without delay, such that $R(s_t, a_t)$ provides informative feedback at each $t$.

In the fully delayed reward setting, the environment provides no feedback during the episode and instead returns a single scalar reward at the end, so we receive only indirect information about $R_{\text{original}}$. In this paper, we assume this setting and focus on turning the delayed reward function into a dense one. The dataset is a collection of episodes, each with a return given at the end of the episode. These returns are the sum of the reward function at each timestep:

$$G_{\text{episode}} = \sum_{t=0}^{T} R_{\text{original}}(s_t, a_t) \tag{4}$$

## 4 Attention-based REward Shaping (ARES)

Figure 1 shows the architecture of ARES. The core idea behind ARES is to generates a dense reward function by manipulating a transformer's attention mechanism. At a high level, ARES trains a transformer model that learns to predict episodic return, and then use that model to assign per-step credit to individual state-action pairs. As the transformer learns to predict episodic return, it learns how the various state-action pairs that make up an episode contribute to the return, and this is reflected in the attention matrix. In the last row of the attention matrix, only the entry of interest is retained, while all other entries are set to 0. The rest of the transformer's calculations then proceed as normal. Algorithm 1 summarize the pseudocode of ARES.

Generally, transformers are trained on some sequence-to-sequence task, where the first token is passed in and used to predict the second, then the first and second tokens are passed in and used to predict the third, and

so on. In our setting, the episodes are made up of one token (a state-action pair) for every timestep. The final token of the episode is the episode's return (summed reward over every timestep). The ARES transformer takes in every token of the episode from the first token to the next-to-last token, with the objective being to predict the very last token, the return of the episode. The transformer architecture, which is inherently able to handle variable length sequences, allows ARES to handle trajectories of different lengths without any special consideration needed. The attention matrix consists of many rows, where the first row is used with the first token to predict the second token, the second row is used with the first and second tokens to predict the third token, and so on. The last row of the attention matrix is used with all the tokens other than the last to predict the last token. In the matrix multiplication of the attention matrix and the values matrix (see Figure 1), the last row of attention is multiplied by the last column of values. If we set all the attention values in the final row to 0 except for some "token of interest," what we get is a sort of lookup from the values matrix, telling us about the contribution of that token of interest to the final reward.

Note that we do *not* set the attention value of the token of interest to 1. If we did so, then we would be losing the information from the attention calculation, and instead be relying only on the values matrix for our shaped rewards. In a perfect world, the values from the value matrix would indeed be perfect representations of the immediate reward for every state-action pair, and this would work. In such a world, each state-action pair would have equal predictive power, and so the numbers in the last row of the attention matrix would all be about equal. But calculating true immediate rewards based off delayed rewards alone is infeasible with current methods, and so in reality the transformer must rely upon the attention values in order to estimate delayed rewards, as a shortcut to lower its training loss.

It is this last point that motivates our unorthodox approach to the delayed-reward problem: we have absolutely no constraint that the shaped rewards must add up to the delayed reward, or even come close! Working free from this constraint is what allows ARES to be such a general algorithm. This is both a strength and a weakness: other papers, such as those working within the potential-based reward shaping framework (A. Y. Ng & Russell., 1999), can provide stronger theoretical guarantees by working with such constraints. The only constraint on the ARES shaped rewards is both indirect and informal: the transformer's training loss converges to some number during training, which means that the values in the last row of the attention matrix and the values in the last column of the values matrix slowly become more accurate (to be precise, they approach values which lead to the transformer having a lower loss on predicting episodic

---

**Algorithm 1** ARES: Attention-based REward Shaping
___
**Require:** Dataset $D$ of episodes of the form $[(s_1, a_1), (s_2, a_2), \ldots, (s_T, a_T), g]$
 1: Initialize transformer model $M$ for training, empty map $S$ for storing shaped rewards
 2: **for all** episodes $e \in D$ **do**
 3:     Input $e$ into $M$ to produce scalar prediction of the episodic return $\hat{g}$
 4:     Compute loss: $\mathcal{L} = \text{MSE}(\hat{g}, g)$
 5:     Update $M$ via gradient descent on $\mathcal{L}$
 6: **end for**
 7: **for all** episodes $e \in D$ **do**
 8:     **for all** timesteps $t$ in $e$ **do**
 9:         Input $e$ into $M$, with an attention mask over all indices other than $t$, to produce $\hat{r}_t$
10:         Store shaped reward: $S[(s_t, a_t)] \leftarrow \hat{r}_t$
11:     **end for**
12: **end for**
13: **return** $S$
14:
15: *Optional: Normalize rewards*
16: $r_{\max} \leftarrow \max\{S[(s, a)] : (s, a) \in S\}$
17: **for all** $(s, a) \in S$ **do**
18:     $S[(s, a)] \leftarrow S[(s, a)]/r_{\max}$
19: **end for**
___

returns). When we extract the shaped rewards, we use those two sets of values, and in a way similar enough to the original calculations so as to benefit from the higher accuracy. If the reader finds this unsatisfying, they are correct: ARES is justified purely by the practical results, which is why we take care to test a range of environments and algorithms.

The result of the ARES algorithm is a map of state-action pairs to rewards. When training, on every timestep, the agent finds the state-action pair in this map closest (usually by a simple linear distance calculation) to the one it currently sees, and uses the reward corresponding to that pair. In our implementation, we use a K-D tree as the map, which leads to only a small amount of overhead (from lookup time) when training the algorithm. We also find more consistent results if we add a simple normalization step, detailed in the pseudocode. For practitioners interesting in using the algorithm on very complicated environments, we do recommend the possible usage of a variance-reduction step, which we did not find necessary for our experiments but is detailed in Appendix B. Note that we would classify ARES as a supervised learning algorithm rather than an inverse RL method. Inverse RL is concerned with learning the reward function that was governing an agent's actions, but in the randomly-acting case, this would not be possible.

## 5 Experiments

### 5.1 Experiment Setup

**Environments and RL Algorithms**: Table 2 summarizes the Gymnasium environments and RL algorithms employed. In each environment, the original dense reward function is replaced with a delayed reward function, $R_{\text{delayed}}$, which assigns a reward of zero at every timestep except the final one, where it returns the total episodic return.

Table 2: RL algorithms and dataset sizes used for each environment.

| Environment | RL Alg. | Random Dataset Sizes | TrainingExpert Dataset Sizes |
|---|---|---|---|
| CliffWalking (modified) | Tabular Q | 100 episodes | 99–136 episodes |
| CartPole | DQN | 200 episodes | 316–499 episodes |
| LunarLander | DQN | 1000 episodes | 219–384 episodes |
| Hopper | SAC, PPO | 2k timesteps | 1M timesteps |
| HalfCheetah | SAC, PPO | 2k timesteps | 5M timesteps |
| Swimmer | SAC, PPO | 20k timesteps | 1M timesteps |
| Walker2d | SAC, PPO | 2k timesteps | 1M timesteps |
| InvertedDoublePendulum | SAC, PPO | 1k timesteps | 1M timesteps |

We select these environments to represent a broad range of settings, covering all four combinations of discrete and continuous states and actions. Specifically, we include five challenging MuJoCo environments (E. Todorov & Tassa., 2012)(DeepMind., 2021) frequently used as benchmarks in this field. For details on all environments and their specific versions, including a modification made to the CliffWalking environment to increase the difficulty of reward shaping, please refer to Appendix B.

**Data:** For each environment, we generate two distinct datasets of episodes. The first, referred to as the **Random** dataset, consists of episodes generated by an agent taking purely random actions and is relatively small. The second, referred to as the **TrainingExpert** dataset, contains episodes produced by an agent trained using the specific reinforcement learning algorithm associated with that environment (e.g., Soft Actor-Critic (SAC) (T. Haarnoja & Levine., 2018) for MuJoCo, as it typically outperforms Proximal Policy Optimization (PPO) (J. Schulman & Klimov., 2017) in these settings). Although the agent eventually learns expert behavior, the majority of episodes in the TrainingExpert dataset are suboptimal. These two datasets represent the spectrum of episode quality that ARES can handle. While ARES can also process datasets comprised solely of expert demonstrations, in the absence of noise, imitation learning methods such as behavior cloning may be more effective. Detailed descriptions of all datasets and the RL algorithm parameters used for expert data generation are provided in Appendix B and Appendix C, respectively. For each environment, we generate 10 independent TrainingExpert datasets and 10 Random datasets.

**Data to shaped rewards:** For each dataset, ARES is trained by iterating over every episode for a fixed number of epochs, performing one gradient descent step per timestep. The training objective is to minimize the mean squared error between the true episodic return and the predicted episodic return. A detailed description of the ARES training parameters for each dataset can be found in Appendix D. Upon completion of training, ARES produces a set of shaped rewards that define a new reward function. In our experiments, we generate one shaped reward function per dataset, totaling 10 shaped reward sets from TrainingExpert datasets and 10 from Random datasets for each environment.

**Shaped rewards for training:** For evaluation, we train an agent on the target environment using the specified RL algorithm and the new reward function generated by ARES. We compare this to agents trained with two baseline reward functions: the **Delayed Rewards** function (serving as a baseline) and the original **Immediate Rewards** function (serving as the gold standard). (Note that the gold standard is not necessarily a realistic measure for achievement in this problem space: normally, as done in a number of the papers we cite, the baseline for comparison is the delayed reward. But we compare against the gold standard of immediate rewards as well, to show the strength of our algorithm.) Importantly, the agent trained with ARES-shaped rewards has no access to the original reward during training; the original reward is only used for evaluation purposes. Each shaped reward set is used for a single training trial, resulting in 10 trials for the Random dataset and 10 for the TrainingExpert dataset per environment. Results from these trials are labeled as TrainingExpert Shaped Rewards and Random Shaped Rewards in the figures. Appendix C provides a detailed description of the RL algorithm parameters used during evaluation.

For evaluation with the three simpler toy environments (CliffWalking, CartPole, and LunarLander), we measure how many agents trained within a certain number of episodes were able to solve the environment. For example, an episode cutoff of 50 means that the agent returns success if the environment is solved within 50 episodes, and returns failure otherwise. On the CliffWalking-m environment this means achieving a reward of $+88$, on CartPole a return of $+500$, and on LunarLander a reward $>= 200$. Since LunarLander is a somewhat "looser" environment than the others (as can be seen by the larger standard deviation) in that it is occasionally possible to achieve this return by accident, we require the agent achieve it 5 times before reporting it as solved. For evaluation with the more complicated MuJoCo environments, every 10,000 steps we report the average return over the past 100 episodes according to the **original** reward function.

**Other algorithms:** It was difficult to choose other algorithms for comparison, since ARES is the only algorithm we know of that is suitable for the most general setting. In the end, we decided on comparison to GAIL (Ho & Ermon., 2016) and to LOGO. We train GAIL on the TrainingExpert dataset. We choose GAIL since it is a commonly-used imitation learning algorithm, and we would like to see if it is better able than ARES to extract information from the TrainingExpert dataset. We also use LOGO trained on the TrainingExpert dataset. We choose LOGO not only because it is a strong algorithm, but because it assumes a lower degree of sparsity than fully delayed. We show how it performs in the fully delayed case alongside ARES, noting that we would expect similar performance when evaluating any algorithm that does not anticipate the fully delayed case. GAIL uses a SAC learner, and LOGO uses a TRPO (a similar algorithm to PPO) learner. Note that performance differs in the MuJoCo environments between PPO and SAC, so we

do not compare to LOGO in the SAC experiments or to GAIL in the PPO experiments. (They are omitted from the toy environments for a similar reason.)

**Architecture** The reader may consult Appendix A for an exhaustive description of all architecture parameters used in our experiments.

## 5.2 Results

**TrainingExpert shaped rewards** As shown in Figures 2 and 3, there are three settings (SAC and PPO HalfCheetah, PPO Swimmer) where the TrainingExpert shaped rewards show lackluster performance. In all ten other settings, they show very strong performance, leading the agent to achieve rewards not too far below the golden standard of the immediate-reward agent. This is more or less an expected result: the TrainingExpert data, while rather noisy and generally only having a small subset of episodes which are truly close to expert-quality, does contain a lot of information, which can be imparted in the ARES shaped rewards. GAIL is not well-equipped to handle the noise and sparsity of expert demonstrations in the TrainingExpert data. Expectedly, the performance of LOGO suffers when the rewards are far more sparse than it was designed for.

**Random shaped rewards** As shown in Figures 2 and 3, there are two settings (SAC Swimmer, PPO Walker2d) in which the delayed rewards perform better than the Random shaped rewards, and four (LunarLander, PPO Swimmer, SAC and PPO HalfCheetah) in which their performance is comparable. In all seven other settings, the Random-data shaped rewards are superior to the delayed rewards.

## 6 Conclusion

While the TrainingExpert shaped reward performance is (perhaps unsurprisingly) superior to the Random shaped reward performance, we would actually consider the latter to be a more remarkable result. The Random data is extremely low quality, and the fact that ARES performs relatively well with it is a happy result. By only taking in a tiny dataset of random data, ARES is generally able to boost the performance of an agent working within a delayed reward environment. This means that without any access to expert data, ARES can often be used to allow an agent to at least partially learn in delayed reward environments, even in ones which seem to be very difficult otherwise: for almost all the environments we examine, there is little to no improvement in the delayed-reward agent's performance even after 1 million timesteps. Note that on two of the eleven Random settings, ARES performed worse than the fully delayed rewards, which shows that ARES can sometimes fail to convert the very low-quality data into usable immediate rewards.

We note **two main limitations of ARES:** one theoretical and one empirical. The first is that ARES has no theoretical guarantee on the quality of the shaped rewards: other algorithms, working in more controlled settings, can give this guarantee while we cannot. The second is the weakness of the results on some environments: for example, the results are quite strong on Walker2d but quite poor on HalfCheetah, despite the fact that these environments both have the exact same dimensionality. All of this is not completely surprising: ARES is meant to be a jack-of-all-trades algorithm, and these two limitations are at least partially attributable to the high difficulty of addressing the problem without making any assumptions or restrictions on the data or setting.

Finally, note that the performance of the ARES shaped rewards is moderately invariant to the RL algorithm used. While in general performance is very different on the MuJoCo environments between SAC and PPO (see the large differences in the Immediate Rewards agents), ARES generally offers a similar kind of increased performance compared relatively between the two algorithms. The exception would be Swimmer, which may be explained by the fact that SAC performs poorly on this environment while PPO performs quite well, in contrast to the other environments. Since the TrainingExpert dataset comes from a SAC agent, this would explain why the TrainingExpert-dataset shaped rewards offer little to no improvement in performance.

In this paper we have introduced ARES, which uses reward shaping to help solve delayed-reward environments. ARES is, to our knowledge, the only algorithm of its kind: it combines a high robustness to episode quality with the ability to work offline and with fully-delayed environments. In addition, it shows com-

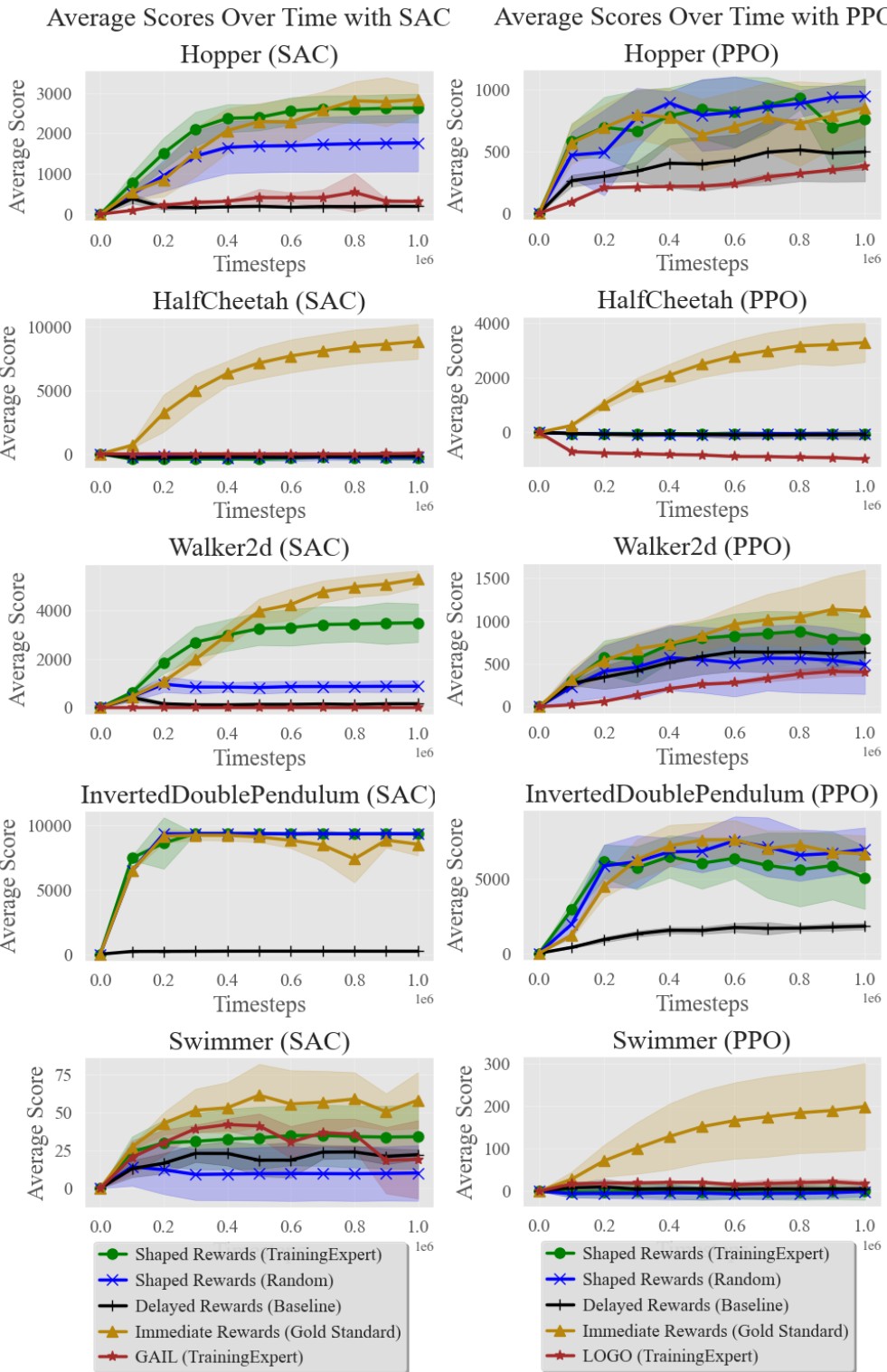

Figure 2: Results on MuJoCo environments. Shaded regions are +/- one standard deviation.

patibility with a wide array of diverse RL algorithms, and it is not restricted to a goal-based RL setting. As a result, it is the most general algorithm to date for solving the sparse and delayed reward problems using shaping. There are other interesting applications for ARES: for example, since we can now generate a

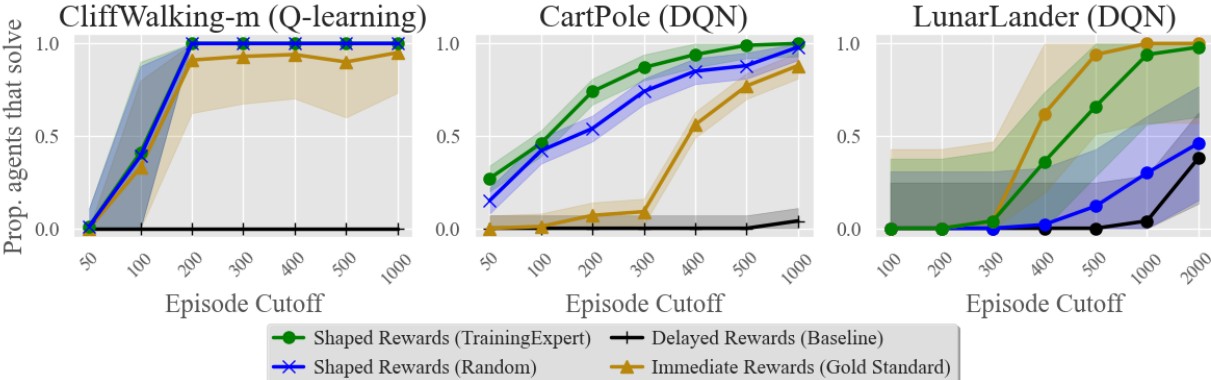

Figure 3: Results on toy environments. Shaded regions are +/- one standard deviation.

shaped reward for every state-action pair in an offline dataset, ARES has the potential to be combined with an offline model-based RL algorithm to learn a delayed-reward environment completely offline. (This was actually the initial purpose for creating ARES: it was only afterwards that it was realized how useful it could be for the delayed reward problem.) Our foremost hope is that ARES proves a genuinely useful method for RL practitioners, and we have taken care to provide a code base that is clear and easily adaptable for new environments. We also hope that others will take up the torch and find improvements to the base algorithm: we feel we have only scratched the surface with the novel technique used for creating shaped rewards.

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

# A  ARES Architecture Parameters

Table 3: Model and training configuration for the ARES transformer

| Component | Details |
| --- | --- |
| Transformer type | GPT with masked causal attention |
| Input dimension | `state_dim + act_dim` |
| Embedding layer | Linear: `token_dim` → `h_dim` |
| Transformer blocks | 1 |
| Block structure | Masked causal attention → MLP [`h_dim` → `internal_embedding` · `h_dim` → GELU → `h_dim`] → residual → LayerNorm → LayerNorm |
| Attention heads | 1 |
| Dropout rate | Same `dropout_rate` used through whole architecture |
| Final projection | Linear(`h_dim` → `token_dim`) → Linear(`token_dim` → 1) |
| Loss function | Mean squared error (MSELoss) |
| Optimizer | AdamW |
| Learning rate | Same `learning_rate` used through whole architecture |
| Max sequence length | 1050 (episodes longer than this are not encountered in our experiments) |
| Tokenization | For every timestep, the state and action are concatenated, to form 1 token for that timestep |
| Training strategy | 1 gradient descent step per episode, repeated `epoch` times with shuffling of the dataset |

Table 4: Model parameters used for transformer architecture

| Hyperparameter | Value |
| --- | --- |
| Token dimension | `state_dim + act_dim` |
| Transformer blocks | 1 |
| Hidden dimension (`h_dim`) | 512 |
| Maximum sequence length (`max_T`) | 1050 |
| Number of attention heads | 1 |
| Dropout rate (`drop_p`) | 0.01 |
| Internal embedding ratio | 8 |

# B  Environments and Datasets

To modify the CliffWalking environment, we add an additional reward of +100 on the final step. While this makes the normal environment easier to solve, it actually makes the delayed reward-shaping problem more difficult. In the normal case, every episode will have a negative return, so the shaped rewards will generally lead to optimal behavior as long as they are all negative; but in the modified case, it is possible to achieve a large positive return, which means that depending on the data it receives the reward-shaping algorithm might be misled into assigning a positive shaped reward to some state other than the final state, which leads to an infinite loop as the agent repeatedly moves into that state. This was a difficult problem to overcome, and we found in our experiments that shaping algorithms which use an LSTM architecture will run into this problem.

We use slightly older versions of the environments for evaluating LOGO and GAIL. For LOGO we use Hopper-v3, HalfCheetah-v3, Swimmer-v3, and Walker2d-v3. For GAIL we use Walker2d-v4. This is not meant to manipulate the results of the baseline algorithms in any way: the only difference in dynamics between any of the older environments and the newer ones is that in the Walker2d-v5 environment, the right foot of the walker has more friction. We use these older environments due to Python compatibility issues between the official LOGO implementation we used, the Python imitation library we used for GAIL, and our own code. Note that in general we prefer to use the latest version of each environment in order to facilitate comparison of our algorithm with other recent papers. We omit InvertedDoublePendulum from our GAIL and LOGO experiments since the older versions of the environment have different dynamics (to be precise, there is one more state in the older versions) and it may be a misleading comparison to evaluate the baselines on the older and more difficult environment and evaluate ARES on the newer and less complex environment.

Table 5: List of environments and their configured versions

| Environment | Version |
|---|---|
| CliffWalking-m | `CliffWalking-v1` with +100 on reaching final state |
| CartPole | `CartPole-v1` |
| LunarLander | `LunarLander-v2` |
| Hopper | `Hopper-v4` |
| HalfCheetah | `HalfCheetah-v4` |
| Swimmer | `Swimmer-v4` |
| Walker2d | `Walker2d-v5` |
| InvertedDoublePendulum | `InvertedDoublePendulum-v5` |

Table 6: TrainingExpert datasets used with toy environments

| Environment | Dataset Description | Dataset Sizes (in Episodes) |
|---|---|---|
| CliffWalking | Datasets of no more than 200 episodes, each containing exactly 1 optimal episode. After the optimal episode is generated, no more episodes are produced. | 99, 103, 100, 110, 112, 108, 122, 124, 125, and 136 episodes |
| CartPole | Datasets of no more than 500 episodes, each containing exactly 1 optimal episode (length of 500). After the optimal episode is generated, no more episodes are produced. | 373, 316, 499, 461, 373, 395, 499, 428, 335, and 496 episodes |
| LunarLander | Datasets of no more than 500 episodes, each containing exactly 5 optimal episodes. After the optimal episodes are generated, no more episodes are produced. | 267, 254, 330, 219, 263, 276, 384, 313, 311, and 262 episodes |

Table 7: Random datasets used with toy environments

| Environment | Dataset Description | Dataset Sizes (in Episodes) |
|---|---|---|
| CliffWalking | Datasets consist of 100 random episodes. Episodes with a reward less than or equal to -2000 are discarded and regenerated. This is only done because randomly generated episodes from this environment can often be unreasonably long (since the only way to end the environment is to hit a very specific goal state). Other papers performing experiments on similar environments prefer to use a semi-trained agent to collect data in order to avoid this problem, but this would violate our assumption of a randomly-acting agent. | 100 episodes |
| CartPole | Datasets consist of 200 random episodes. | 200 episodes |
| LunarLander | Datasets consist of 1000 random episodes. | 1000 episodes |

Table 8: TrainingExpert datasets used with MuJoCo environments

| Environment | Dataset Description | Dataset Sizes (in Timesteps) |
| --- | --- | --- |
| Hopper | 1,000,000 timesteps generated by a SAC agent. | 1,000,000 timesteps |
| Swimmer | 1,000,000 timesteps generated by a SAC agent. | 1,000,000 timesteps |
| Walker2d | 1,000,000 timesteps generated by a SAC agent. | 1,000,000 timesteps |
| InvertedDoublePendulum | 1,000,000 timesteps generated by a SAC agent. | 1,000,000 timesteps |
| HalfCheetah | 5,000,000 timesteps generated by a SAC agent. | 5,000,000 timesteps |

Table 9: Random datasets used with MuJoCo environments

| Environment | Dataset Description | Dataset Sizes (in Timesteps) |
| --- | --- | --- |
| Hopper | Random dataset generated by executing a randomly-acting policy in the Hopper-v4 environment. | 2,000 timesteps |
| Swimmer | Random dataset generated by executing a randomly-acting policy in the Swimmer-v4 environment. Note that Swimmer does not have a truncation condition like the other MuJoCo environments, but always runs for 1,000 timesteps on each episode, so if we generated 2,000 timesteps as with the other environments, we would only have two episodes. | 20,000 timesteps |
| HalfCheetah | Random dataset generated by executing a randomly-acting policy in the HalfCheetah-v4 environment. | 2,000 timesteps |
| Walker2d | Random dataset generated by executing a randomly-acting policy in the Walker2d-v5 environment. | 2,000 timesteps |
| InvertedDoublePendulum | Random dataset generated by executing a randomly-acting policy in the InvertedDoublePendulum-v5 environment. | 1,000 timesteps |

## C  RL Algorithm Parameters

The same RL algorithm parameters are used for both training and evaluation.

Table 10: Tabular Q-learning parameters

| Parameter | Value used | Description |
| --- | --- | --- |
| epsilon | 0.9 | Initial exploration rate for $\epsilon$-greedy policy |
| Decay rate for epsilon | 0.99 | Every step epsilon is multiplied by this value |
| gamma | 0.99 | Discount factor |
| learning_rate | 0.99 | Step size for Q-value updates |

Note that for SAC on the HalfCheetah environment, we use a lower learning rate of 1e-5.

Table 11: SAC parameters (Stable-Baselines3 defaults)

| Parameter | Value used (default unless otherwise specified) | Description |
|---|---|---|
| policy | MlpPolicy (not default) | Policy class, which must be specified (no default value) |
| learning_rate | 0.0003 | Optimizer learning rate |
| buffer_size | 1,000,000 | Size of the replay buffer |
| learning_starts | 100 | Timesteps before learning starts |
| batch_size | 256 | Size of minibatches sampled from the replay buffer |
| tau | 0.005 | Target smoothing coefficient (soft update) |
| gamma | 0.99 | Discount factor |
| train_freq | 1 | Frequency of training updates |
| gradient_steps | 1 | Number of gradient steps per training iteration |
| action_noise | None | Optional noise for exploration |
| replay_buffer_class | None | Custom replay buffer class (optional) |
| replay_buffer_kwargs | None | Keyword args for custom replay buffer |
| optimize_memory_usage | False | Save memory by not storing next observations |
| ent_coef | 'auto' | Entropy coefficient or 'auto' for adaptive |
| target_update_interval | 1 | Frequency of target network updates |
| target_entropy | 'auto' | Target entropy for entropy tuning |
| use_sde | False | Whether to use State-Dependent Exploration |
| sde_sample_freq | -1 | Frequency to resample exploration noise |
| use_sde_at_warmup | False | Use SDE during warmup phase |
| policy_kwargs | None | Additional kwargs for the policy network |
| seed | None | Random seed |

Table 12: PPO parameters (Stable-Baselines3 custom configuration)

| Parameter | Value used (default unless otherwise specified) | Description |
|---|---|---|
| policy | MlpPolicy (not default) | Policy class, which must be specified (no default value) |
| learning_rate | 9.80828e-05 (not default) | Optimizer learning rate |
| n_steps | 512 (not default) | Number of steps to run for each environment per update |
| batch_size | 32 (not default) | Minibatch size |
| n_epochs | 5 | Number of epochs when optimizing the surrogate loss |
| gamma | 0.999 (not default) | Discount factor |
| gae_lambda | 0.99 | Factor for trade-off of bias vs variance for GAE |
| clip_range | 0.2 | Clipping parameter for PPO |
| ent_coef | 0.00229519 (not default) | Entropy coefficient for exploration |
| vf_coef | 0.835671 (not default) | Value function loss coefficient |
| max_grad_norm | 0.7 (not default) | Maximum gradient norm for clipping |
| use_sde | False | Whether to use State-Dependent Exploration |
| sde_sample_freq | -1 | Frequency for resampling SDE noise |
| target_kl | None | Target KL divergence threshold |
| policy_kwargs | {log_std_init=-2, ortho_init=False, activation_fn=nn.ReLU, net_arch={pi: [256, 256], vf: [256, 256]}} (not default) | Additional arguments for policy network architecture |
| seed | None | Random seed |

Table 13: DQN parameters for CartPole

| Parameter | Value used | Description |
|---|---|---|
| epsilon_start | 0.9 | Initial $\epsilon$ value for $\epsilon$-greedy policy |
| epsilon_end | 0.01 | Minimum $\epsilon$ value |
| epsilon_decay | 1000 (exp) and 0.999 (alt) | Used in both exponential and multiplicative decay formulas |
| gamma | 0.99 | Discount factor |
| learning_rate | 0.0001 | Optimizer learning rate |
| batch_size | 128 | Size of sampled minibatch from replay buffer |
| buffer_size | 10,000 | Maximum replay buffer capacity |
| tau | 0.005 | Target network soft update coefficient |
| optimizer | AdamW (AMSGrad) | Optimizer with AMSGrad variant |
| Network architecture: | [128, 128] | Two fully connected layers with 128 units each |

Table 14: DQN parameters for LunarLander

| Parameter | Value used | Description |
|---|---|---|
| epsilon_start | 0.9 | Initial $\epsilon$ value for $\epsilon$-greedy policy |
| epsilon_end | 0.01 | Minimum $\epsilon$ value |
| epsilon_decay | 0.995 | Multiplicative decay factor per episode |
| gamma | 0.99 | Discount factor |
| learning_rate | 0.0005 | Optimizer learning rate |
| batch_size | 64 | Size of sampled minibatch from replay buffer |
| buffer_size | 100,000 | Maximum replay buffer capacity |
| tau | 0.001 | Target network soft update coefficient |
| update_every | 4 | Frequency (in steps) of learning updates |
| optimizer | Adam | Optimizer for Q-network parameters |
| Network architecture | [64, 64] | Two fully connected layers with 64 units each |

# D  ARES Training Parameters

Table 15: Training parameters for expert datasets

| Environment | Number of Training Epochs |
| --- | --- |
| Hopper | 10,000 steps |
| HalfCheetah | 10,000 steps |
| Swimmer | 10,000 steps |
| Walker2d | 10,000 steps |
| InvertedDoublePendulum | 10,000 steps |

Table 16: Training parameters for random datasets

| Environment | Number of Training Epochs |
| --- | --- |
| Hopper | 5,000 steps |
| HalfCheetah | 1,500 steps |
| Swimmer | 5,000 steps |
| Walker2d | 5,000 steps |
| InvertedDoublePendulum | 5,000 steps |

All other parameters used in the architecture may be found under Appendix A.

# E  ARES hyperparameters

Table 17: Distance metric hyperparameters

| Parameter | Value / Options | Description |
|---|---|---|
| k_neighbors | Integer (e.g., 5, 10) | Number of nearest neighbors used for matching in the KDTree |
| p_state | 1 = Manhattan, 2 = Euclidean | Distance metric used for computing similarity between states |
| p_action | 1 = Manhattan, 2 = Euclidean | Distance metric used for computing similarity between actions |
| rounding | 0 = none, 1 = nearest 0.1, 2 = nearest 0.5, 3 = nearest integer | Rounding strategy applied during preprocessing of the state-action pairs from the map |

A KD-tree is used as the map of state-action pairs to shaped rewards. This vastly reduces the overhead from lookup times that would otherwise be incurred by, say, using a normal Python map/dictionary.

We use the set of values for the above parameters $(5, 1, 1, 0)$ for working with the TrainingExpert dataset and $(5, 2, 2, 0)$ for working with the Random dataset. The only exception is the HalfCheetah environment, where we use $(1, 2, 2, 3)$ as the parameters for both datasets. We found that the agent had better performance on the HalfCheetah environment with this particular set. While the default values of this set of hyperparameters usually lead to good results, one way to boost the performance of ARES is to try and optimize over them.

## F  Guide for Practitioners

**Compute Resources**

All of our experiments were run on a CPU. In particular, they were run on an HPC using Intel Xeon Silver 4110 CPUs with a speed of 2.10GHz. Training an agent on the toy environments is generally minimal in compute requirement. Training an agent on the MuJoCo environments using SAC and PPO took roughly one to six hours per agent using 4 parallelized environments. LOGO took about ten minutes per agent using only one environment, so the authors of the algorithm (whose implementation we use) clearly did an excellent job optimizing performance. GAIL took about ten minutes per agent, since we use 100 parallelized environments. Generating the shaped rewards generally takes one to two hours for the smaller Random datasets, and roughly 10 to 20 hours for the very large TrainingExpert datasets, when using 10,000 epochs for training the transformer. This is mostly because the transformer does not use batching and training it does not use multithreading. With a KD-tree, the overhead added by our algorithm is minimal.

**Tip 1: Working with Discrete Environments**

When working with an environment that has discrete states and actions, we recommend replacing the first PyTorch Linear layer of the transformer with a PyTorch Embedding Layer. We do not do this in the paper, but it will greatly lower training loss and time on discrete environments.

**Tip 2: Variance Reduction Step**

---
**Algorithm 2** ARES: Attention-based REward Shaping
---
*Optional: Smoothing based on proximity:* **Input:** distance metric distance(), distance parameter $\epsilon$ each $(s_i, a_i) \in S$ each $(s_j, a_j) \in S'$, where $(s_i, a_i) \neq (s_j, a_j)$ distance$((s_i, a_i), (s_j, a_j)) < \epsilon$ $r_{\text{avg}} \leftarrow \frac{1}{2}(S[(s_i, a_i)] + S[(s_j, a_j)])$ Create new merged entry $(s^*, a^*)$ from an average of $(s_i, a_i)$ and $(s_j, a_j)$ Remove $(s_i, a_i)$ and $(s_j, a_j)$ from $S$ Add $(s^*, a^*) \mapsto r_{\text{avg}}$ to $S$
---

This merging step can drastically reduce the (already small) overhead from lookups, while actually increasing performance by reducing variance (since we would expect very similar state-action pairs to have very similar rewards). Included in the code base is multithreaded Python code that performs this merging: `merge_rewards_parallel.py`.

**Tip 3: Modifying the Internals of the Transformer**

We use the same architecture and embedding dimensions for all of our experiments, from CliffWalking (48 discrete states, 4 continuous actions) to HalfCheetah (17 continuous states, 6 continuous actions). We found that modification of the architecture and embedding dimensions had very little effect, especially compared to the effects of modifying the parameters of the RL algorithms.

**Tip 4: Increasing Performance**

If the practitioner is running ARES on a particular environment without success, there are two key suggestions we would make: first, encourage exploration by modifying the underlying RL algorithm, in order to help the agent avoid getting stuck in local optima. For SAC, there are four prominent parameters, which are easily modfied by passing them into the `generate_SAC_inferred_output_parameters.py` python file.

Second, the practitioner may find it beneficial to perform a parameter search over the distance metric hyperparameters. Combining these two steps by performing a large parameter search would be the best solution, though a potentially expensive one. The default parameters should be sufficient for a wide variety of environments, but when working with environments that are more complicated than the ones we examine in this paper, these steps are likely to help the practitioner achieve strong results with ARES.

**Tip 5: Working with non-fully-delayed environments**

We might have some non-fully-delayed environment, where the reward is given, say, every 100 timesteps rather than at the end of each episode. There is a simple way to use ARES with such an environment, by only modifying the form of the data. Recall that usually, we assume the dataset is a series of episodes, each

with a reward only given at the end. If in our dataset, every episode instead had a reward given every $T$ timesteps (note that $T$ can be variable both within and between episodes), we can simply break down each episode, treating each period of delay ($T$ timesteps) and then reward as a new episode. The algorithm and architecture are completely compatible with this change, and even if $T$ is not constant: this is what makes a transformer architecture particularly useful for this problem.

**Tip 6: Non-Markovian Reward Functions and Positional Encoding**

If the practitioner is working with an environment that has a non-Markovian reward function, then they may be interested in expanding the transformer and using positional encoding, which allows the transformer to consider the order of the state-action pairs in the episode, rather than seeing all the state-action pairs as a jumbled and unordered mess. A variable for positional encoding is already set up in the transformer code; it is set to 0, since we do not use it for our experiments. The original transformer code (linked in the ARES transformer code) shows how a simple positional encoding is done.

