# OpenReview forum: "Attention-Based Reward Shaping for Sparse and Delayed Rewards"
_TMLR — Rejected by TMLR_

### Review · Reviewer_xCCz · 2025-12-06

**Summary Of Contributions:**

## Paper summary:
This paper addresses the challenge of delayed and sparse rewards, which often hinders learning efficiency and stability in real-world applications. The authors propose ARES, an attention-based reward-shaping method designed to transform sparse or delayed reward signals into dense ones. The method is trained offline, requires only non-expert trajectories and returns, and aims to generate shaped rewards that can be used with any RL algorithm. Empirically, the paper claims to demonstrate that ARES improves learning in several environments and claims relative invariance to the underlying RL algorithm.
## Paper contributions:
1. Proposing Attention-based reward shaping (ARES) method to generate more dense rewards in sparse and delayed reward environments, using non-expert trajectories and training offline.
2. Demonstrates ARES effectiveness in a wide range of RL environments, while been relatively invariant to the RL algorithms.

**Additional Comments:**

## Strengths:
1. **Clear acknowledgment of limitations**: Authors explicitly recognize environments where ARES fails. This transparency strengthens the credibility of the paper.
2. **Reproducibility**: The paper includes comprehensive code and datasets, facilitating validation and further research.
3. **Practical relevance**: Reward shaping for sparse/delayed environments is an important and timely problem, and framing the solution as an offline method is appealing for real-world scenarios.
## Weaknesses:
1. **Lack of Theoretical Foundations**: The paper presents an entirely empirical contribution without theoretical justification, authors acknowledge that. Since the central claims rely on empirical performance, a theoretical discussion, or at least hypotheses grounded in reward hypothesis [1], would strengthen confidence in the method's general applicability.
2. **Internal Conceptual Contradictions**: There is an inconsistency between:
   - Contribution 2: “ARES is moderately invariant to RL algorithms”.
    - Conclusion section: “ARES is not entirely invariant to RL algorithms”.
    - These terms (“moderate,” “invariant”) should be clearly defined and consistently used.
3. Please check why claims are not suported with evidance Section.
## Questions to Authors:
1. What is the exact mathematical distinction between sparse and delayed rewards?
2. Does “offline” refer to training solely from collected trajectories?
3. How does replacing environment reward “with more informative values” align with or contradict the reward hypothesis [1]?
4. How does ARES generalize to unobserved state–action pairs far from training data (out-of-distribution)?
5. How do the shaped rewards relate to the original cumulative reward?
6. How does the method handle varied trajectory lengths?
7. Are hyperparameters tuned per environment and per algorithm?
8. How do the authors define “data quality” in the Random dataset?

## References:
[1] Richard Sutton and Andrew Barto. Reinforcement learning: An introduction. 2018.

[2] Moore AW. Efficient memory-based learning for robot control. University of Cambridge, Computer Laboratory; 1990.

**Audience:**

Yes

**Audience Explanation:**

Reward shaping for sparse and delayed environments is an important problem, and framing the solution as an offline method is appealing for real-world scenarios.

**Claims And Evidence:**

No

**Claims Explanation:**

1. **Missing Explanations for When/Why the Method Works**: Although ARES performs well in certain settings, the paper does not analyze why it works in those environments and why it underperforms in others. For example, Figure 2 shows that in the Swimmer environment (SAC & PPO) and in the Walker2d environment (PPO), the delayed reward baseline outperforms ARES (Random). Therefore, training an agent with the original sparse/delayed environment rewards outperforms ARES with random trajectories, which provides a counter example for **Contribution 1**.
2. **Unrealistic Reward Replacement Assumption**: The method replaces all immediate rewards with zero, except for the episodic return. This assumption is unrealistic for many environments, for example: Negative rewards for undesirable actions shaping penalties until a goal is reached (e.g., MountainCar [2]).Thus, the proposed framework may not generalize to those settings.
3. **Limited Algorithmic Diversity**: Although the paper claims that ARES is “moderate invariant to the RL algorithm,” **(Contribution 2)**, experiments include only PPO and SAC. These are popular algorithms, but it is well-known that RL algorithms vary widely in their sensitivity to sparse rewards. The evidence provided is insufficient to support claims about algorithmic invariance.
4. **The discount factor** asummped to be euqal to one, $\gamma=1$, as in the problem setting Section; however, most RL settings assume $\gamma \in [0, 1)$, making the reward function constructed by ARES invalid.

**Requested Changes:**

1. Additional Experiments:
    - Analyze Success and Failure Modes. Add experiments explaining why ARES succeeds in certain environments/algorithms and fails in others. This is essential given the method’s empirical nature.
    - Dataset Size and state–action coverage: Provide a histogram plot to show dataset representation. Conduct ablations varying dataset size (e.g., 1%, 10%, 50%) to test robustness.
    - Expand RL Algorithm Evaluation: Include more algorithms, such as DQN (discrete actions), TRPO, A3C, DDPG, or others, to justify Contribution 2, claims of invariance to RL algorithm.
2. Clarifications and Missing Components:
    - Add equation labels throughout the paper.
    - Add: “Contribution 2 is justified purely by empirical results.”
    - Add missing citations for: “PPO” and “SAC” on page 7, under “Data” Subsection, and “GAIL” and “LOGO” on page 8, in the “other algorithms” Subsection.
    - Clarify the mathematical definitions of “sparse” and “delayed” rewards in Section 3.
    - Define what the authors mean by “offline training” (e.g., learning from a fixed replay buffer).
    - Explain how ARES handles trajectories of different lengths.
    - Explain how attention weights form a unified reward function.
    - Provide assumptions (e.g., deterministic and stationary environment).
    - Clarify the distinction between expert and non-expert data.
    - Explain how the discount factor affects attention and shapes reward consistency.
    - Describe the computational resources required to train large transformers.
    - Explain what is meant by the dataset “quality” and how it was measured.
    - Add “to the best of our knowledge” to “all previously proposed algorithms fail to meet the criteria”.
3. Adding Missing Related Work:
    - Add the related work: O. Marom and B. Rosman, “Belief Reward Shaping in Reinforcement Learning”, AAAI, 2018
4. Fix Typos:
    - Page 4 and 5, $\sum_{t=0}^{T}$ to match the trajectory definition $(s_0, a_a, s_1, a_1, ..., s_{T}, a_T)$.

---

> ### Author Response · Authors · 2025-12-14
> **Response to reviewer xCCz [1]**
>
> # Reply to “Are claims supported...”
>
> **1. Missing Explanations for When/Why the Method Works**
>
> The reviewer is correct that there should be more analysis of success and failure settings, and that the language regarding the contributions should be made clearer. We have tried to be careful in our claims: we avoid claiming that ARES unilaterally improves performance, only that it generally does so, with explicit caveats in the paper.
>
> There are two reasons for the failure settings, which we discuss below and can include in the paper if they address the reviewer’s concern.
>
> The first reason is MuJoCo termination conditions. For Swimmer, the criticism is certainly clearest: the agent using low-quality data does not outperform delayed rewards. A similar issue appears in HalfCheetah. The cause lies in environment specifications. In many MuJoCo tasks (e.g., Hopper, Walker2d), the simulation terminates if the robot enters an unhealthy state (e.g., falling). In Swimmer and HalfCheetah, however, episodes terminate only at a fixed horizon of 1000 timesteps. This significantly increases return variance, especially in random datasets, and adds substantial noise into the shaped rewards.
>
> The second reason is the choice of RL algorithm. For Walker2d with PPO, the agent performs similarly to or worse than delayed rewards despite having a termination condition. In this setting, both delayed rewards and low-quality shaped rewards exhibit unusually high variance, as reflected in the shaded regions. This is due to PPO performing poorly on this environment, as seen by the gap between PPO and SAC. More generally, when the underlying RL algorithm is poorly suited to an environment (e.g., SAC on Swimmer or PPO on Walker2d), ARES cannot provide substantial improvement and may even reduce performance. In practice, this can often be mitigated by selecting a more appropriate RL algorithm.
>
> These issues are distinct but overlap in Swimmer. Since SAC is used to generate expert-level data, and SAC performs poorly on Swimmer, the expert dataset is not substantially better than random data, which explains the weak performance of expert-shaped rewards.
>
> If the reviewer feels that answering “Yes” to this question requires the authors to reduce the strength of the claims made in the paper, that is also a reasonable compromise.
>
> **2. Unrealistic Reward Replacement Assumption**
>
> The reviewer raises two related concerns: the fully delayed reward setting examined in the paper, and settings where rewards include per-step shaping components.
>
> We evaluate the most difficult setting: rewards fully delayed until episode end. In standard MuJoCo environments, rewards are informative at every timestep and consist of both positive and negative components. We mask this per-step reward and instead provide only the summed return at the end. While this is not intended to model a typical real-world reward signal directly, it serves as a useful proxy for settings where evaluation is episodic and difficult to decompose manually. For example, a system may receive a final evaluation based on multiple factors accumulated over time, where hand-designed per-step shaping is impractical.
>
> ARES is easily extensible to less difficult cases, including environments with per-step control costs or rewards delayed but not until episode end. As noted in the appendix (“Guide for Practitioners”), such cases can be handled by treating each reward-delimited segment as an episode, which generally simplifies reward decomposition relative to the fully delayed setting studied here.
>
> **3. Limited Algorithmic Diversity**
>
> We evaluate four algorithms: tabular Q-learning, DQN, SAC, and PPO. SAC and PPO were chosen as two widely used modern actor–critic methods representing different branches of the RL algorithm family. Closely related algorithms (e.g., TRPO, A3C, DDPG) are not expected to behave too differently from their exemplars. We are happy to include additional algorithms if the reviewer has specific suggestions.
>
> **4. The Discount Factor**
>
> The reviewer may be referring to the summed reward given at the final timestep, denoted \( G_{\text{episode}} \). The discount factor is a hyperparameter used by RL algorithms during optimization and does not affect the ARES algorithm itself. (For completeness, we use a discount factor of 0.99 for most experiments, and 0.999 in one case, as reported in the appendix.)

---

> > ### Author Response · Authors · 2025-12-14
> > **Response to reviewer xCCz [2]**
> >
> > # Requested Changes
> >
> > **Additional Experiments**
> >
> > We understand the causes of success and failure modes discussed above and plan to incorporate this analysis into the paper. If the reviewer has specific experiments they believe would be informative, we are happy to add them.
> >
> > Regarding dataset size and state–action coverage, we intentionally test two extremes of data quality: small random datasets with minimal coverage, and large datasets generated by agents that eventually achieve high return. The random datasets have extremely limited coverage (e.g., in Hopper, the agent typically falls immediately), yet ARES can sometimes generalize despite this limitation. We chose these extremes to demonstrate that comprehensive state–action coverage is not strictly required for ARES to be effective.
> >
> > **Clarifications and Missing Components**
> >
> > - Add equation labels throughout the paper. **(Added)**
> > - Clarify that Contribution 2 is empirically justified. **(Added)**
> > - Add missing citations for PPO, SAC, and GAIL. **(Added)**
> > - Clarify mathematical definitions of sparse and delayed rewards. **(Added)**
> > - Define “offline training” as learning from a fixed dataset. **(Added)**
> > - Explain handling of variable-length trajectories. **(Added)**
> > - Add “to the best of our knowledge” to related-work claims. **(Added)**
> > - Add missing related work. **(Added)**
> > - Fix typos. **(Fixed)**
> >
> > # Weaknesses
> >
> > **Lack of Theoretical Foundations.**
> > The paper presents an empirical reward-shaping method without formal guarantees, which we explicitly acknowledge. Our contribution emphasizes rigorous empirical evaluation across challenging settings rather than theoretical optimality guarantees.
> >
> > **Internal Conceptual Contradictions.**
> > Language has been revised for consistency and clarity.
> >
> > # Questions to Authors
> >
> > **What is the distinction between sparse and delayed rewards?**
> > Mathematical definitions are now provided in Section 3.
> >
> > **Does “offline” refer to training solely from collected trajectories?**
> > Yes (we have added a note on this to the paper).
> >
> > **How does replacing environment rewards with more informative values relate to the reward hypothesis?**
> > As mentioned above, if the reviewer can explain what they mean with regards to connecting our work to the reward hypothesis, we can give an answer. In the settings we examine, we are replacing the zero-information immediate rewards with rewards that do give information about how “good” the current state-action is, in terms of maximizing the return (summed reward) as per the reward hypothesis.
> >
> > For example, in the normal Hopper environment, the agent receives, among other things, a positive reward for moving to the right; this encourages the agent to move to the right, because doing so leads to maximizing the return. When the reward is fully delayed, the agent cannot learn this behavior, because it lacks the information it needs to learn how to maximize the return (despite the fact that optimal behavior is the same in both the normal immediate-reward setting and the delayed-reward setting). But if we replace the fully-delayed reward function (0 on every step but the last) with the ARES rewards, the agent can once again learn to move to the right, because we once again have a reward function that is informative. We could add this explanation to the paper to make “more informative values” more clear, if it meets the reviewer’s approval.
> >
> >
> > **How does ARES generalize to unobserved state–action pairs?**
> > Rewards are assigned via nearest-neighbor lookup in state–action space. Despite limited coverage in random datasets, ARES can often generalize effectively, as shown empirically.
> >
> > **How do shaped rewards relate to original cumulative reward?**
> > There is no constraint that shaped rewards sum to the delayed reward, which enables generality but explains certain failure modes.
> >
> > **How are varied trajectory lengths handled?**
> > The transformer architecture naturally supports variable-length sequences, and we have added a note on this to the algorithm description.
> >
> > **Are hyperparameters tuned per environment and per algorithm?**
> > Hyperparameters were not tuned in any exhaustive way. We use consistent hyperparameters across most of the environments to ensure fair comparison.
> >
> > **How is data quality defined?**
> > Data quality refers to the source of trajectories: purely random actions versus trajectories generated by an agent that eventually achieves high return.

---

### Review · Reviewer_9XKr · 2025-12-06

**Summary Of Contributions:**

This paper proposes using attention-based mechanism to solve the delayed problem. In such setup, the algorithm looks at the weights calculated by attention layer to see how much of the states contribute to the resulting delayed reward. By doing so, the algorithm can provide temporally dense reward, highlighting which state was more useful towards the end reward and calculate dense reward along the way.

The major strength of this paper that it attempts to solve one of the major problem in RL ; delayed reward. Using the attention mechanism is intuitive and an straightforward attempt as well.

One of the most major weakness of the paper is that it considers all contributions to be in the same direction. The attention mechanism serves as analytical tool to see which states contributed more towards the end reward. However, attention alone cannot tell the direction of the influence, meaning that a state with big attention value could have adversarily / positively affected the end reward, but attention alone cannot tell the direction of the influence.

Another weakness of the paper, as the authors mentioned, is the lack of theoretical guarantee on the method. While this can be substituted by empirical results, the paper could use more experiments to empirically support a concept that has yet been proven theoretically.

**Additional Comments:**

N/A

**Audience:**

Yes

**Audience Explanation:**

Providing dense reward to a spare or delayed reward problem is a commonly reoccurring challenge in RL.

**Claims And Evidence:**

No

**Claims Explanation:**

While the idea is interesting, I am not quite sure if is possible to verify whether an state with high attention value has positively / negatively affected the results, as attention can calculate the magnitude of the importance, not the direction of the effect. Therefore it is difficult to tell if the algorithm is capable of calculating the dense reward properly.

**Requested Changes:**

Verify that the attention based algorithm can tell the direction as well as the magnitude on the effect of the states on final reward.
Either a theoretical proof on prediction accuracy and relevance / more empirical results suggesting that the idea applies to a variety of problems in CS.

Also, verify whether attention can be used to predict a state's affect towards final reward, not just in terms of magnitude.

---

> ### Author Response · Authors · 2025-12-14
> **Response to reviewer 9XKr**
>
> Note that the attention weights themselves do not form the reward function, although they are a part of it. We do use the unmodified attention weight for each timestep when performing our “lookup” into the transformer, which as shown in Figure 1 is done by a series of steps that end up with essentially grabbing a particular entry from the values matrix. The attention weights themselves are only a small part of the process, and mostly serve to help us have an easy way to do this lookup. But as noted in section 4, rather than setting them to 1 each time, we do intentionally choose to keep the attention values as they’ve been set to by the transformer:
>
> “Note that we do not set the attention value of the token of interest to 1. If we did so, then we would be losing the information from the attention calculation, and instead be relying only on the values matrix for our shaped rewards. In a perfect world, the values from the value matrix would indeed be perfect representations of the immediate reward for every state-action pair, and this would work. In such a world, each state-action pair would have equal predictive power, and so the numbers in the last row of the attention matrix would all be about equal. But calculating true immediate rewards based off delayed rewards alone is infeasible with current methods, and so in reality the transformer must rely upon the attention values in order to estimate delayed rewards, as a shortcut to lower its training loss.”
>
> Since what we find in the value matrix could be positive or negative, ARES has no problem dealing with rewards of various magnitudes and directions. A paper which does use the attention weights for a reward function would be the ABC paper that we discuss in Section 2. They use the attention weights directly, and while their method is novel and interesting, this means that there are a number of limitations, the foremost being that the method can only handle rewards all of one sign (i.e. positive or negative, not both). It sounds like that might be more along the lines of what the reviewer is thinking of, and while a generally valid concern, it is not applicable to our algorithm.
>
> Furthermore, one of the pain points we had very early on during our research was that the attention weights are not always reflective of numerical impact when the transformer is trained to predict delayed rewards given episodes. It was clear to us that attention weights alone cannot be relied upon to form a coherent reward function, since as we mention above they are first and foremost a heuristic for the transformer.
>
> Normally, the MuJoCo robotic environments have an immediate reward which is given every step. This reward has multiple components: there is some positive component regarding success (for example, a positive reward given for moving to the right), and a negative component, which is a control cost that penalizes actions. Higher magnitude actions incur a higher cost. This difficulty means any algorithm like ours has to “disentangle” the positive from the negative somehow. This is very difficult: in one of the papers under our “Related Work” section, the authors had to actually manually integrate the MuJoCo control cost equation into their algorithm, because they found it so difficult to derive offline. ARES does not require anything like this: as we said, it is inherently able to deal with varying magnitudes and directions.
>
> (That’s also why we ablate the CliffWalking environment by adding a +100 reward on reaching the goal: normally CliffWalking simply has -1 on every step and -100 for falling off the cliff, and we wanted to obfuscate this setting by adding a large positive reward.)
>
> So to sum up: other than CartPole, all the environments we test (CliffWalking, LunarLander, and MuJoCo) mix positive and negative rewards. And because the attention weights are only a part of the final shaped reward calculation, the algorithm can handle rewards of different directions. This seems to be the reviewer’s main concern: if there are any others that we can address, please let us know.
>
> If the above discussion is to the reviewer’s satisfaction, we can include it in the paper. If the reviewer is still unconvinced and feels that in order to answer “Yes” to the question regarding evidence, the authors should reduce the claims made in the paper, that is also a reasonable compromise.

---

### Review · Reviewer_w5wg · 2025-12-07

**Summary Of Contributions:**

The paper introduces ARES (Attention-based Reward Shaping), a new method for transforming sparse and fully delayed reward signals into dense per-timestep rewards using a transformer based architecture. The key idea is to train a transformer to predict the episodic return from a sequence of state–action tokens and then extract per-step shaped rewards by manipulating the final attention layer. This produces a dense reward function without requiring expert demonstrations or online interaction. The main contributions are:
A general reward shaping algorithm for sparse and fully delayed rewards. ARES is designed to satisfy all of the following simultaneously: (a) works in fully delayed reward settings, (b) operates entirely offline, (c) handles non-expert and even random trajectories, (d) is applicable to general RL environments (not limited to goal-based RL). The paper claims that no previous method meets all these conditions at once.


ARES leverages the transformer’s attention matrix to infer which timestep contributes most to the final return, producing shaped rewards for each state–action pair. This approach imposes no requirement that shaped rewards sum to the actual return. The paper provides extensive empirical evaluation across diverse environments and RL algorithms. Experiments span classic control (CliffWalking, CartPole, LunarLander) and several MuJoCo benchmarks (HalfCheetah, Walker2d, Hopper, Swimmer, InvertedDoublePendulum), using both SAC and PPO. With TrainingExpert datasets, ARES often approaches the performance of immediate-reward agents, except in a few environments. With Random datasets, ARES often outperforms delayed-reward baselines, showing robustness to low-quality data. Comparisons to GAIL and LOGO highlight ARES's advantages in settings with extreme reward sparsity. The paper provides detailed architecture parameters, hyperparameters, and practitioner tips, aiming to make ARES easy to apply and tune.


Strengths:
- The proposed approach is one of the few reward-shaping methods to work simultaneously in offline, fully delayed, non-expert, and general RL settings. This is a notable contribution because existing methods typically fail one or more of these criteria.
- Leveraging the last-row attention vector to assign credit is innovative. The idea that reward shaping need not preserve return sums allows the method to work in settings where frameworks like potential-based shaping are too restrictive.
- The ability to produce meaningful dense rewards—even with random action data—is impressive and potentially valuable for real-world domains where high-quality trajectories are scarce.
- Experiments span classic control (CliffWalking, CartPole, LunarLander) and several MuJoCo benchmarks (HalfCheetah, Walker2d, Hopper, Swimmer, InvertedDoublePendulum), using both SAC and PPO.
- The paper provides detailed architecture parameters, hyperparameters, and practitioner tips, aiming to make ARES easy to apply and tune.





Limitations:
- Lack of theoretical guarantees. ARES does not ensure policy invariance or return-preserving transformations. The shaped rewards need not sum to the original return. The key mechanism—reinterpreting attention weights as credit—is interesting but somewhat heuristic.
- It is not clear why attention aligns with causal credit and when it might fail. How does the learned reward relate to the true reward structure? Are the results sensitive to architecture choices?
- The method approximates unseen state–action pairs by nearest-neighbor lookup in high-dimensional space. This may introduce sensitivity to dimensionality, or propagate noise from reward estimates, and depends heavily on preprocessing and distance metrics.
- The paper compares to LOGO and GAIL, but not to other transformer-based temporal credit assignment models (e.g., RRD, predictive coding methods, SECRET).

**Audience:**

Yes

**Audience Explanation:**

TMLR serves a broad ML community including researchers focused on reinforcement learning, sequence modeling, offline learning, credit assignment, and representation learning. This paper intersects several of these areas. Reinforcement Learning with Sparse/Delayed Rewards is a long-standing, technically challenging problem (e.g., RUDDER, RRD, LOGO). ARES directly addresses fully delayed reward settings, which remain under-explored due to difficulty. There is rising interest in using sequence models for RL (Decision Transformers, trajectory models, credit assignment via attention). Offline RL is an active research area with challenges around low-quality data and lack of immediate feedback. ARES’s ability to operate with small, noisy, purely offline datasets may draw interest from applied practitioners. The empirical findings contribute meaningful knowledge about the limits of attention-based credit assignment. The practical implementation details make the method usable by practitioners.

**Broader Impact Concerns:**

The authors could include a Broader Impact Statement that comments on:
- Safety risks of using an unverified dense reward in real-world settings
- Misaligned incentives and reward hacking
- Risks of relying on low-quality offline data
- Potential misuse in safety-critical or ethically sensitive domains
- Interpretability in attention-based reward shaping

**Claims And Evidence:**

Yes

**Claims Explanation:**

ARES is broadly applicable. A systematic comparison table (Table 1) where each prior method is assessed against the criteria: handling delayed rewards, whether it works offline, using non-expert data, and if it is general (not goal-based) RL. The experiments suggest that ARES improves learning in delayed reward settings across many environments and is moderately invariant to the choice of RL algorithm (SAC vs PPO). They also support the claim that ARES can generate meaningful rewards even using small, random datasets.

The claim that attention-based reward extraction is effective is empirically demonstrated but it is not clear why attention reflects meaningful credit. There are no ablation studies to show: what happens if attention is not altered, whether specific heads/rows matter and whether the results are sensitive to transformer architecture.

**Requested Changes:**

Suggested Changes (not critical)

1. Provide deeper analysis of failure cases (e.g., HalfCheetah, Swimmer) to obtain plausible explanations (e.g., reward landscape, transformer limitations, dataset mismatch).

2. Add ablations isolating key components of ARES, e.g. no-attention variant, uniform final-row mask, varying transformer depth/heads, normalization.

3. Add interpretability analysis of learned shaped rewards and discuss sensitivity to hyperparameters, especially distance metrics in KD-tree matching.

4. Add a discussion on limitations related to high-dimensional state–action spaces.

5. Include a broader discussion of potential real-world applications.

---

> ### Author Response · Authors · 2025-12-14
> **Response to reviewer w5wg**
>
> # Limitations:
>
> Note that the attention weights themselves do not form the reward function, although they are a part of it. We do use the unmodified attention weight for each timestep when performing our “lookup” into the transformer, which as shown in Figure 1 is done by a series of steps that end up with essentially grabbing a particular entry from the values matrix.
>
> The new shaped rewards are generally very different from the true immediate reward of the given environment. This is not necessarily surprising: reconstructing the true immediate reward is probably not something which is possible using current mathematical or computational techniques, which is why we chose to work without the constraint of requiring the learned rewards to equate to the delayed rewards in any meaningful way.
>
> Regarding architecture: one nice thing about the algorithm is that the architecture has little to no impact on the shaped rewards, as long as the delayed rewards are predicted with a relatively high degree of accuracy (which is usually not an issue).
>
> # The method approximates...
>
> This is indeed a possibility. The state-action coverage in, for example, the Hopper random dataset is abysmal. Perhaps the most surprising thing about ARES is that it is able to sometimes overcome this extremely low state-action coverage and create a set of shaped rewards that generalize to the entire environment space.
>
> # The paper compares...
>
> We would have liked to test against some of the other methods, but none of these methods are suitable for the same setting that ARES is tested on. That being said, many of these algorithms are certainly very good for the constrained use cases they are designed for, and doubtlessly some would outperform ARES on those particular cases.
>
> # Provide deeper analysis...
>
> For Swimmer, the criticism is certainly clearest: the agent using low-quality data does not outperform delayed rewards. A similar issue appears in HalfCheetah. The cause lies in environment specifications. In many MuJoCo tasks (e.g., Hopper, Walker2d), the simulation terminates if the robot enters an unhealthy state (e.g., falling). In Swimmer and HalfCheetah, however, episodes terminate only at a fixed horizon of 1000 timesteps. This significantly increases return variance, especially in random datasets, and adds substantial noise into the shaped rewards.
>
> For Walker2d with PPO, the agent performs similarly to or worse than delayed rewards despite having a termination condition. In this setting, both delayed rewards and low-quality shaped rewards exhibit unusually high variance, as reflected in the shaded regions. This is due to PPO performing poorly on this environment, as seen by the gap between PPO and SAC. More generally, when the underlying RL algorithm is poorly suited to an environment (e.g., SAC on Swimmer or PPO on Walker2d), ARES cannot provide substantial improvement and may even reduce performance. In practice, this can often be mitigated by selecting a more appropriate RL algorithm.
>
> These issues are distinct but overlap in Swimmer. Since SAC is used to generate expert-level data, and SAC performs poorly on Swimmer, the expert dataset is not substantially better than random data, which explains the weak performance of expert-shaped rewards.
>
> # Add ablations...
>
> We have tried varying the transformer architecture (mostly in order to make it quicker to train) and normalization. A no-attention variant would not be possible since the attention weights themselves are used to index into the value matrix. Varying the transformer architecture and normalization both had little to no impact on the shaped rewards.
>
> # Add interpretability analysis...
>
> The learned shaped rewards are generally only meaningful in relation to one another, since they come from a different embedding space than the original rewards. The algorithm is indeed slightly sensitive to the hyperparameters of the KD-tree matching, but the hyperparameters that we ended up choosing were universally good for almost all of the environments.
>
> # Add a discussion...
>
> Based on our experiments, it seems that factors other than the dimensionality of the state-action spaces have the most impact on ARES. We can include this point in the paper.
>
> # Include a broader discussion of potential real-world applications & Broader Impact Concerns:
> ARES can be used to create a shaped reward for any possible delayed-reward scenario: all the practitioner has to provide is some sort of delayed reward function to measure the desirability of various episodes. We would hope that any practitioner using ARES in a real-world setting sees the flaws of the algorithm, as demonstrated in our experiments, and understands that it is not infallible. Part of our hope for this work is that it is expanded upon by others and made more robust and extensible.
> If the above points meet the reviewer’s satisfaction, we can include them in the conclusion of the paper as well.

---

### Comment · Action_Editor_qRRv · 2025-12-27
**Question about "reward sparsity"**

Dear reviewers and authors,

I would like to raise a main concern regarding how reward sparsity is framed and instantiated in the paper, and invite both reviewers and authors to share their perspective on this point. I acknowledge that in the original submission the terms sparse and delayed reward were largely conflated, and that the revised version makes a clear effort to disentangle these notions.

However, the core difficulty usually associated with sparse-reward problems in RL, namely the difficulty of encountering informative rewards under feasible exploration policies, and the consequent need for structured or directed exploration, is not addressed by the current definition of sparse reward, which focuses mainly on the temporal sparsity or delay of the reward signal.

While the experimental evaluation is certainly extensive and informative, I only see one task (CliffWalking) that plausibly corresponds to an original sparse-reward setting. The discussion for this task, however, primarily emphasizes credit-assignment pathologies (such as reward misattribution and looping) rather than exploration-related challenges. Even for CliffWalking, the dataset construction for the Random setting essentially avoids this by applying acceptance-rejection sampling based on return. This appears to bias the data toward trajectories that terminate more quickly and receive informative rewards, thereby reducing the exploration difficulty that typically motivates sparse-reward benchmarks.

I would be interested to hear your thoughts on whether this distinction between delayed reward, sparse reward, and exploration difficulty is adequately addressed by the paper, and whether it should affect how the contribution is positioned or evaluated.

Thanks again

---

> ### Author Response · Authors · 2025-12-27
> **Response to reviewer qRRV**
>
> We thank the reviewer for sharing their concern. To make the distinction between the terms "delayed" and "sparse" defined in the introduction clearer, we have also added a mathematical distinction, as suggested by one of the reviewers.
>
> The reviewer’s concern seems to be mainly centered around the exploration problem, which is quite understandable. It’s an interesting point, because generally speaking, the discussion around reward shaping involves some sort of focus on exploration. And despite being a reward shaping algorithm, we abstract the exploration problem to whatever RL algorithm is in use (i.e. the entropy parameter). Why do we do this? There are a few reasons, a main one being that trying to tackle exploration alongside the delayed-reward problem is not feasible. Like most of the related works, we treat these as two completely separate problems, and whether that is entirely correct of us to do is rather messy and quite debatable.
>
> The other main reason why we are able to abstract away this problem is because of the strength of ARES in generalizing to unseen state-action spaces. This is something other reviewers were also very curious about, and is probably the most surprising result of the algorithm. For example, on the MuJoCo settings, recall that the Random dataset is nothing more than a set of episodes where the robot takes completely random actions until it falls over. There’s a great deal of noise in the dataset because of the MuJoCo control cost and because there is no immediate reward, but somewhere underneath all the noise is the obfuscated information that moving to the right leads to a positive reward. Using only this very low-quality data, ARES can sometimes, but not always, output a new immediate-reward function that gives this information to the agent much more clearly than the original delayed-reward function.
>
> Also, note that we work entirely offline when reconstructing the rewards. If we have some environment like the original MountainCar, for example, where there’s only a reward of 1 for reaching the summit and 0 otherwise, and the dataset has no examples of the agent reaching the summit, then it would not be possible to reconstruct this reward from the offline dataset, as there are no examples of it. This is not a problem in the environments we chose, but there are certainly settings where ARES and other similar algorithms would not be applicable because of the issue. To put it formally: ARES is primarily a method for addressing delayed/sparse reward problems and the temporal credit assignment problem in offline settings, rather than problems where exploration is the main issue.
>
> In the delayed-reward environments we explore, exploration is not really the problem: the problem is that the delayed-reward functions do not give sufficient information to the RL algorithm to allow the agent to learn (i.e. increase the amount of positive reward it receives) very well. That’s why a reward-shaping algorithm that encourages exploration does not help to solve these environments. But you might say that ARES encourages the agent to explore areas of the state-action space with high reward in the same way that the immediate-reward function does, since both give informative feedback and are much better at informing the agent what to explore and what not to explore as compared to the delayed-reward function. In this way you could view exploration as an integral part of the ARES shaped rewards, though that is not the way we frame the problem.
>
> Regarding the environments we chose: as the reviewer notes, most of the environments are originally dense-reward settings that we modified to have a delayed reward. This is the same procedure followed in many of the related works we cite (which modify the MuJoCo environments to have sparse rewards), so we hope it does not seem unorthodox or haphazard. We don’t worry about exploration much except on CliffWalking, where as you point out, we have to throw out many of the trajectories since a randomly-acting agent usually generates absurdly long episodes. The CliffWalking environment is not a strong sparse-exploration benchmark because of this, and in the MuJoCo environments we don’t do anything like this, since as the reviewer notes it does indeed bias the dataset towards being easier to work with.
>
> We are nearing the character limit, but if the reviewer has any other concerns (such as the applicability of ARES to varying degrees of reward sparsity, which we discuss in another response), please let us know. We can include a more formal version of this exploration discussion in the paper, if the reviewer feels it would be helpful to do so.

---

### Decision · Action_Editor_qRRv · 2026-01-25

**Recommendation:** Reject

**Additional Comments:**

I believe the paper could be significantly strengthened by addressing the following points:
- Clearly distinguish between delayed reward, sparse reward, and exploration-related difficulty, and position the contribution accordingly.
- Provide a deeper analysis of cases where the method underperforms, including settings in which delayed-reward baselines outperform ARES.
- Strengthen or appropriately moderate claims of robustness and algorithm invariance, either through broader algorithmic coverage or clearer empirical justification.
- Clarify key modeling assumptions, including the role of discounting and the interpretation of attention-based credit assignment. Additional ablations and diagnostic experiments that directly test the necessity and reliability of the attention mechanism would further strengthen the work.

**Audience:**

Yes

**Audience Explanation:**

Yes. Reward shaping for sparse or delayed rewards, and the use of sequence models/attention for credit assignment, are topics of broad interest in the RL and ML communities. The idea of producing dense shaped rewards from offline trajectories and episodic returns is potentially useful, and the experimental exploration is informative even when results are mixed.

**Claims And Evidence:**

No

**Claims Explanation:**

The paper makes central claims regarding robustness, generality, algorithm invariance, and effectiveness with low-quality data. While there is empirical evidence showing that the proposed method can improve learning in some delayed-reward settings, the current problem formulation, benchmark selection, and experimental results do not justify the breadth and generality of the claims being made. Performance is mixed across environments and algorithms, and several counterexamples remain insufficiently explained, making it unclear when and why the method should be expected to work reliably.

**Resubmission Of Major Revision:**

The authors may consider submitting a major revision at a later time.